# Outpatient Management of Aural Fullness: A Retrospective Case Series of 100 Patients with Cerumen Impaction, Keratosis Obturans, and External Auditory Canal Cholesteatoma

**DOI:** 10.3390/life15121936

**Published:** 2025-12-18

**Authors:** Giovanni Motta, Domenico Testa, Giuseppe Barba, Rosa Grassia, Francesco Chiari, Arianna Di Stadio, Giuseppe Tortoriello

**Affiliations:** 1Otolaryngology Head and Neck Surgery Unit, “Azienda Ospedaliera di Rilievo Nazionale dei Colli, Ospedale Monaldi”, 80131 Naples, Italy; 2ENT Unit, Department of Mental, Physical Health and Preventive Medicine, University of Campania “Luigi Vanvitelli”, 80131 Naples, Italy; 3Otolaryngology, Head and Neck Unit, “Santo Spirito” Hospital, 65124 Pescara, Italy

**Keywords:** cerumen impaction (CI), aural fullness, otalgia, otorrhea, external auditory canal cholesteatoma (EACC), keratosis obturans (KO)

## Abstract

**Background:** Aural fullness is a common symptom in routine otolaryngological practice. Although it is most commonly attributed to cerumen impaction, other, less frequent conditions may present similar symptoms and are often initially misdiagnosed as cerumen. These include keratosis obturans (KO) and external auditory canal cholesteatoma (EACC). Accurate differentiation among these entities is crucial for appropriate management. These distinctions are crucial for appropriate management. **Methods:** We retrospectively reviewed 100 patients who presented with a chief complaint of aural fullness from 2021 to 2025. All patients underwent microscopic and/or micro/endoscopic otologic evaluation and were subsequently treated with outpatient otologic procedures. These procedures ranged from simple cerumen removal for CI cases to aural toilettage of the external auditory canal for KO and initial debridement attempts for EACC. **Results:** Among 100 patients, 87 were diagnosed with CI, 10 were diagnosed with KO, and 3 were diagnosed with EACC. In 97 patients, outpatient microscopic management was effective and successful, leading to the complete removal of the underlying condition and resolution of the ear fullness. In the remaining 3 cases—all diagnosed with EACC—microscopic outpatient debridement was not sufficient. These patients were subsequently scheduled for surgical intervention following audiological and temporal bone CT evaluations. **Conclusions:** Our findings confirm that CI is the most frequent cause of aural fullness and that microscopic outpatient removal represents an excellent standard of care. However, clinicians should remain aware that KO and EACC may present similar symptoms. Their management is often more complex, potentially requiring multiple sessions and, in the case of EACC, can necessitate surgical intervention. Accurate diagnosis is, therefore, essential to ensure effective and appropriate treatment.

## 1. Introduction

Aural fullness, whether isolated or associated with other otologic symptoms such as pain, itching and otorrhea, constitutes one of the most common reasons for otolaryngology consultation. In the vast majority of cases, this symptom is caused by CI, which typically produces a sensation of fullness accompanied by only mild discomfort or minor associated symptoms [1,2].

However, other, albeit rarer, otologic conditions exist that may present similar obstructive symptoms but require substantially different management. These conditions, often misdiagnosed as refractory or difficult-to-remove sub-occlusive cerumen, include KO and EACC [3,4,5]. The ability to accurately recognize and differentiate between these entities is paramount for appropriate management.

Distinguishing between these conditions has historically been challenging. For many decades, KO and EACC were considered variants of the same underlying disease. This diagnostic conflation stemmed from their superficial clinical similarities, primarily presenting as an obstructive condition within the external auditory canal. It was only subsequent to improved knowledge that these two conditions were definitely recognized as separate pathological entities, each possessing distinct features and pathophysiology [3,4,5,6,7,8,9,10,11]. KO is characterized by the accumulation of a dense, exfoliated keratin plug that, through chronic pressure, causes a concentric widening and remodeling of the bony external ear canal. In contrast, EACC is an inflammatory and erosive process that typically results in a focal lesion leading to osteo-erosion of the underlying bone [3,4,5,6,7,8,9,10,11,12,13,14,15,16,17,18]. This distinction is clinically critical, as EACC inherently involves progressive bone erosion and, depending on the extent of the lesion and potential involvement of surrounding structures, may require surgical intervention to achieve definitive eradication and to prevent complications [3,4,5,6,7,8,9,10,11,12,13,14,15,16,17,18].

Given the significant disparity in treatment and the ongoing clinical challenge of differentiating between these conditions, particularly KO from EACC, focused clinical data are needed. This paper presents a retrospective, single-center, single-surgeon case series of 100 patients presenting with aural fullness, with or without additional otologic manifestations, who ultimately received a diagnosis of cerumen impaction, KO, or EACC. The goal of this study is to describe these representative causes of aural fullness encountered in routine ENT practice and to outline their distinguishing features and management strategies, emphasizing the importance of accurate differential diagnosis.

## 2. Materials and Methods

This retrospective, multi-center, single-surgeon case series involved 100 consecutive patients (73 males, 27 females), included between 2021 and 2025. The mean age of the cohort in this study was 63 years (range 22–89 years). All patients presented to the outpatient otolaryngology clinic with a chief complaint of aural fullness. The study strictly focused on the three most common and/or representative obstructive causes of aural fullness within the external auditory canal (EAC). Etiologies regarding middle or inner ear pathology (Eustachian tube dysfunction, middle ear effusion and sudden hearing loss) were excluded from our investigation. The definitive diagnosis for this cohort included cerumen impaction (87 patients), KO (10 patients), and EACC (3 patients). At presentation, 59 patients reported mild otalgia in the absence of otorrhea. Baseline characteristics revealed that 70 patients were current or past smokers, 15 were affected by chronic rhinosinusitis with or without nasal polyposis, and 8 suffered from atopic dermatitis. All participants underwent a comprehensive otologic examination using either otomicroscopy or oto-microendoscopy. Management consisted of single (87 patients) or repeated (10 patients) outpatient micro-otologic procedures performed in the outpatient setting. No external auditory canal bleeding was observed in 87 cases. Conversely, minor external auditory canal bleeding was noted in 13 cases. Among these 13 cases, the bleeding pattern was classified as focal in 3 debridement procedures and concentric in the remaining 10 cases. Only 3 patients, following initial outpatient micro-otologic debridement, were further evaluated with pure-tone audiometry and a non-contrast temporal bone CT scan. The primary data collected were post-procedural, focusing on the patient’s subjective improvement in hearing. A structured follow-up for repeat outpatient toilettage of the external auditory canal was performed in 10 patients.

## 3. Results

A total of 100 patients presenting with aural fullness between 2021 and 2025 underwent outpatient cerumen removal, external auditory canal (EAC) toilettage, and/or debridement procedures. The overall success rate, defined as resolution of aural fullness and subjective improvement in hearing, was 97%. Specifically, success was achieved in a single session for 87 patients with cerumen impaction. In contrast, all 10 patients with keratosis obturans (KO) required multiple toileting sessions for successful resolution. Of these 10 patients, 5 were seen twice within a 3-month period, while the other 5 were seen three times within a 3-month period to perform periodic aural toilettes. In the remaining 3% of cases, all diagnosed with external auditory canal cholesteatoma (EACC), outpatient otologic debridement was insufficient. During micro-endoscopic removal, no external auditory canal bleeding was observed in 87 cases of cerumen impaction. Conversely, minor external auditory canal bleeding was noted in 13 cases. Among these 13 cases, the bleeding pattern was classified as focal in 3 debridement procedures (EACC) and concentric in the remaining 10 cases (KO). In the 3 patients diagnosed with EACC, the procedure failed to completely remove the pathology or improve the patient’s acoustic perception due to significant osteo-erosion and the extension of the cholesteatomatous disease into the mastoid and middle ear cavity, which was present in all three cases. Consequently, these three patients were further evaluated with pure-tone audiometry and non-contrast temporal bone CT scans to assess hearing status and the extent of the cholesteatomatous disease. These examinations confirmed moderate conductive hearing loss in all cases, while the CT scan demonstrated the extension of the pathology into the mastoid and middle ear. Based on these findings, surgical intervention was proposed to these patients. Additionally, none of the three patients diagnosed with EACC had undergone surgery at the time of manuscript preparation. Surgical intervention was recommended based on audiometric and radiologic evidence of osteo-erosion and disease extension; however, because definitive operative treatment had not yet been performed, detailed surgical outcomes fall outside the scope of this study, which focuses primarily on outpatient diagnostic differentiation and conservative management. A dedicated table summarizing the number of patients diagnosed with CI, KO, and EACC is provided (Table 1).

## 4. Discussion

Aural fullness, whether isolated or associated with other otologic symptoms such as pain, itching, or otorrhea, is among the most frequent reasons for otolaryngology consultation. In most cases, this symptom results from cerumen impaction, which typically causes a sensation of fullness without significant pain or with only mild discomfort [1,2]. However, other less frequent but more severe conditions can also cause aural fullness and may be accompanied by pain, itching, and otorrhea. These conditions, including keratosis obturans (KO) and external auditory canal cholesteatoma (EACC), can be easily mistaken for simple cerumen impaction upon initial otoscopic examination, particularly by less-experienced ENT clinicians. It is therefore crucial to correctly identify and differentiate these entities, as their management differs substantially [3,4,5,6,7,8,9,10,11]. The key clinical differences between KO and simple cerumen impaction involve the presence of symptoms indicative of external auditory canal remodeling and neovascularization. KO is characterized by a dense plug of well-organized, ribbon-like keratin that exerts significant kinetic energy and pressure. This leads to the expansion and remodeling of the bony external auditory canal, a process which frequently presents as otalgia. Impacted cerumen, while often mistaken for KO, does not typically cause this underlying bony change and expansion. Moreover, KO is a hallmark of bleeding upon removal. The bleeding is attributed to the chronic irritation and expansion of the external auditory canal, which induces the formation of fragile new blood capillaries on the plug’s periphery, which is characterized by a silvery-white matrix. These capillaries rupture when the plug is extracted. Conversely, bleeding in cases of simple impacted cerumen is a rare finding and is most often iatrogenic [1,2,10,11,12,13]. While conservative microscopic or micro-endoscopic outpatient removal is highly effective for cerumen impaction and periodic aural toileting is a successful strategy for KO, these same outpatient strategies, including surgical debridement, are often ineffective and insufficient for EACC [1,2,3,4,5,6,7,8,9,10,11,12,13,14,15,16,17,18,19,20,21,22,23,24,25,26,27,28,29,30,31]. This is due to the inherent characteristics of EACC, which can cause osteo-erosion and, at times, extend into adjacent structures such as the mastoid and middle ear cavity [4,5,6,7,8,9,10,11,12,13,14,15,16,17,18,19,20,21,22,23,24,25,26,27,28,29,30,31]. The distinction between these otologic entities is not always straightforward, even for experienced otologists. Historically, KO and EACC were often considered a single entity or used interchangeably. Only with advancements in clinical and pathological understanding were KO and EACC recognized as distinct entities with different natural histories and management requirements. Early cases described in the 19th century by Toynbee (1850) and Scholefield (1893) likely included both conditions without a clear distinction [3,4,5,6,7,8,9,10,11,12,13,14,15,16,17,18,19,20,21,22,23,24,25,26,27,28,29,30,31]. While both pathologies involve the accumulation of desquamated keratin, they differ in location, spread, and effect on the underlying bone structure. KO involves an abnormal accumulation of keratin that forms a large plug encapsulating the entire circumference of the external auditory canal. Its effect on the bone is expansion. EACC is typically a localized inflammatory and infectious process that leads to infection of the surrounding bone. This results in focal erosion of the underlying bone rather than concentric expansion [3,4,5,6,7,8,9,10,11,12,13,14,15,16,17,18,19,20,21,22,23,24,25,26,27,28,29,30,31]. To avoid misdiagnosis, and based on the experience of this case series, the approach to these conditions must involve oto-microscopic or oto-micro-endoscopic evaluation to assess the feasibility of conservative management. The findings from this case series reinforce the previously mentioned distinctions. Cerumen impaction accounted for 87% of cases and was uniformly resolved with rapid outpatient micro-otologic removal, leading to a notable subjective improvement in hearing and a resolution of the sensation of fullness. In 10% of cases, the condition was diagnosed as KO rather than simple cerumen. The first four KO cases, managed by the lead author during the initial three years of residency, required prolonged outpatient removal times of approximately 35 min. At that stage, due to a lack of experience, the author initially believed this to be severely impacted cerumen. The realization that it was KO came only after several minutes of work under the microscope, even because of excessive bleeding during removal. The subsequent six KO cases were managed more quickly, in about 15 min per session. However, to achieve full symptom resolution and subjective hearing improvement, all ten patients with KO required at least two toileting sessions, as the condition tended to recur at irregular intervals. All ten KO patients shared common distinguishing features on oto-micro-endoscopy: a keratin-like material tenaciously adhered to the walls of the EAC and was significantly more difficult to remove than common cerumen. This material was typically located throughout the canal but was most prominent in its deep, bony portion, near the tympanic membrane (which occasionally appeared inflamed). It also caused a concentric enlargement of the canal, but crucially, without signs of osteonecrosis or bony erosion—a key differentiator from EACC [3,4,5,6,7,8,9,10,11,12,13,14,15,16,17,18,19,20,21,22,23,24,25,26,27,28,29,30,31]. In the three EACC cases within this series, the attempted debridement was unsuccessful due to the presence of bony erosion. This necessitated further evaluation with pure-tone audiometry and non-contrast temporal bone CT scans. The audiometric findings documented a moderate conductive hearing loss in all cases, while the CT scans confirmed the presence of an erosive cholesteatomatous pathology, which had extended into the mastoid and into the middle ear cavity in all instances. Therefore, outpatient surgical debridement cannot be considered an adequate management strategy for such cases. These patients were thus informed about the necessity and feasibility of surgical intervention in the operating room. In contrast to KO, for which repeated outpatient toilettage was effective in all cases, EACC could not be adequately managed with conservative debridement. This difference reflects the distinct underlying pathophysiology: KO causes circumferential canal widening without bone destruction, whereas EACC produces focal osteo-erosion and may extend into the mastoid or middle ear, making outpatient treatment impossible. Although surgical intervention was recommended for all three EACC cases based on clinical, audiometric, and radiological findings, none had yet undergone surgery at the time of manuscript preparation. For this reason, operative techniques and outcomes are not detailed, as the purpose of this study is to emphasize outpatient diagnostic evaluation rather than surgical management. A differential among these three conditions, including symptomatology, otological findings, and management, and also based on clinical experience from this paper, is described in Table 2. The significance of this case series lies not only in providing a descriptive account of both common and rare otologic conditions but also in highlighting the critical importance of using a microscope or both a microscope and an endoscope in routine otologic outpatient practice. This approach is essential to correctly identify and differentiate entities like KO and EACC, which are often misdiagnosed as simple impacted cerumen, particularly in the case of KO [3,4,5,6,7,8,9,10,11,12,13]. Only consistent micro- or micro-endoscopic practice provides the necessary tools and skills for accurate diagnosis and the selection of the most appropriate therapeutic strategies for each of these distinct conditions, from the simplest to the most complex.

## 5. Conclusions

Aural fullness, with or without accompanying otologic symptoms, is a common reason for ENT consultation. Most cases are attributable to cerumen impaction, which can be reliably and effectively managed with microscopic or micro-endoscopic removal in the outpatient setting. However, a subset of patients presents with less common conditions, such as keratosis obturans (KO) and external auditory canal cholesteatoma (EACC), for which routine outpatient toilettage may not be sufficient. In this case series, KO frequently requires multiple sessions of external auditory canal toilettage to achieve symptom resolution, underscoring the chronic and insidious nature of the condition. Conversely, EACC associated with osteo-erosion or extension into adjacent structures could not be adequately treated with outpatient debridement alone and typically necessitates surgical management in the operating room. These findings emphasize the importance of accurate differential diagnosis and the essential role of microscopic and endoscopic evaluation in guiding appropriate treatment strategies. While the majority of cases are caused by cerumen impaction, which is consistently and successfully managed with micro-endoscopic removal in an outpatient setting, a subset of patients presents with rarer conditions. In these cases, such as Keratosis Obturans (KO) and External Auditory Canal Carcinoma (EACC), the success of simple micro-endoscopic toilettage is not always guaranteed. Based on our experience with this case series, a conservative outpatient approach involving multiple external auditory canal toilettes may be necessary for KO. Conversely, EACC with osteo-erosion and extension to adjacent structures often necessitates surgical strategies in the operating room, as simple surgical debridement is neither sufficient to resolve nor control the disease.

## Figures and Tables

**Table 1 life-15-01936-t001:** Distribution of patients diagnosed with CI, KO, and EACC.

Diagnosis	Number of Patients (*n*)	Percentage (%)
Cerumen Impaction	87	87%
Keratosis Obturans (KO)	10	10%
External Auditory Canal Cholesteatoma (EACC)	3	3%

**Table 2 life-15-01936-t002:** Clinical features, otoscopic findings and management strategies for CI, KO, and EACC.

	Cerumen Impaction	Keratosis Obturans (KO)	External Auditory Canal Cholesteatoma (EACC)
**Symptoms**	Aural fullness, mild discomfort.	Aural fullness, pain, and recurrence of symptoms.	Aural fullness, pain, otorrhea, and sometimes itching.
**Otoscopic** **Findings**	Visible cerumen plug; no bony erosion.	Keratinous material tenaciously adhered to canal walls, diffuse concentric widening of the canal; no bony erosion.	Focal lesion, clear signs of focal bony erosion and osteonecrosis.
**Management**	Single outpatient microscopic removal.	Multiple, periodic outpatient toileting sessions.	Outpatient debridement is often ineffective; it requires a CT scan and probable surgical intervention.

## Data Availability

Data are available upon reasonable request to the corresponding authors.

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
