# Peer review of "Outpatient Management of Aural Fullness: A Retrospective Case Series of 100 Patients with Cerumen Impaction, Keratosis Obturans, and External Auditory Canal Cholesteatoma"

_life, 2025, doi:10.3390/life15121936_

Round 1

Reviewer 1 Report

Comments and Suggestions for Authors

This retrospective study discusses management in 100 patients presenting with aural fullness. The objective of this study is unclear. 

Introduction: The authors have not mentioned other etiologies of aural fullness, for example, eustachian tube dysfunction, otomycosis, middle ear effusion,  and sudden hearing loss.

Methods: Why do authors mention "recruiting" patients between 2021 to 2025 when the study was retrospective? Please mention the inclusion and exclusion criteria for the study. How were patients "recruited"? 

Results: What outcome measures did the authors evaluate? What statistical methods were used for the analysis of data? 

Discussion: Authors should discuss all the etiologies of aural fullness. What are the unique reflections and considerations alluded to by the authors in the title? 

Comments on the Quality of English Language

English-language editing and careful revision of grammar and typos are necessary. For example, line 25 "outpatient 1ntological procedures"; line 27 "toilettages". 

Author Response

We appreciate the Reviewer's insightful comments, which point to a necessary clarification regarding the scope and specific objective of our study.

Regarding the exclusion of other etiologies of aural fullness:

We acknowledge that other etiologies of aural fullness exist (e.g., Eustachian tube dysfunction, otitis media with effusion, sudden sensorineural hearing loss). However, these conditions were intentionally excluded from our investigation.

Our study was strictly focused on obstructive causes of aural fullness within the external auditory canal (EAC). We systematically excluded all etiologies originating from middle or inner ear pathology (e.g., Eustachian tube dysfunction, otitis media with effusion, sudden sensorineural hearing loss).

This selection was based on their prevalence and critical interrelationship in clinical practice. We focused on the three most representative and relevant obstructive conditions: cerumen impaction, keratosis obturans, and external auditory canal cholesteatoma. Conditions that historically have posed problems of differential diagnosis between them.

Cerumen impaction, being the most frequent cause of EAC obstruction, often requires differentiation from keratosis obturans. In turn, keratosis obturans must be accurately distinguished from the more clinically significant EAC cholesteatoma. Our aim was to clarify the diagnostic pathway among these specific, obstructive entities. While cerumen impaction is the most common etiology, the core of our work lies in delineating the crucial differential diagnosis between these three conditions to ensure prompt and precise management for these obstructive causes of aural fullness in the EAC.

In conclusion, the objective of our study was to highlight three of the most common and clinically representative causes of aural fullness encountered in routine daily otolaryngological practice and to delineate their distinct management strategies.

While cerumen impaction is typically the most frequent etiology, it is essential for clinicians to be aware of and consider two other significant obstructive conditions affecting the external auditory canal: Keratosis Obturans (KO) and External Auditory Canal Cholesteatoma (EACC).

Historically, the distinction between KO and EACC has been challenging, and this difficulty may persist if these entities are not correctly recognized and differentiated. Our findings underscore the need for a precise identification when dealing with patients presenting with obstructive aural fullness related to such causes because the management of these conditions is different.

Given that this study is a descriptive retrospective case series, conventional criteria for statistical analysis and formal study appraisal are not fully applicable. Due to this reason, title has been changed to reflect more accurately the design of our study.

The manuscript has been amended to reflect more clearly the precise scope and objective.

Thank you for your comments aiming at improving our manuscript

Reviewer 2 Report

Comments and Suggestions for Authors

Thank you very much for the opportunity to review this interesting manuscript. The authors present a valuable case series based on the extensive clinical experience of a senior otologist, compiling numerous patients presenting with aural fullness and analyzing them through that expert lens. The manuscript is clearly written and overall very easy to follow.

Below, I would like to offer several candid questions and suggestions, particularly from the perspective of younger otologists or trainees who may not yet have subspecialty-level experience in otology.

1. On the clinical findings and imaging

From the overall content, the authors appear to suggest that among patients presenting with aural fullness, cerumen impaction is the most common cause, followed by keratosis obturans (KO) and external auditory canal cholesteatoma (EACC), and that distinguishing KO from EACC is clinically important.

In lines 95–96, the authors state that “All participants underwent a comprehensive otologic examination using either otomicroscopy or oto-microendoscopy.

Do the authors have photographic documentation of typical otoscopic findings for KO and EACC? Demonstrating the similarities in external auditory canal appearance between these conditions—while also highlighting the differences, such as the pattern of bleeding after intervention noted in lines 123–124—would be extremely educational for less-experienced clinicians.

If available, representative images comparing KO and EACC would greatly strengthen the paper. Additionally, comparative CT images illustrating the key radiologic distinctions would be highly beneficial.

2. On patient background factors (lines 93–95)

The manuscript notes that 70 patients were current or past smokers, 15 had chronic rhinosinusitis with or without nasal polyposis, and 8 had atopic dermatitis.

Did any of these factors show a potential association with the clinical presentation or diagnosis?

If so, it may be helpful to comment briefly on their possible implications in the Discussion section. If feasible, providing some statistical analysis related to these background variables could further enhance the rigor of the manuscript.

3. On Table 1 and potential statistical analysis

Table 1 is very informative in summarizing the clinical signs associated with cerumen impaction, KO, and EACC. As the authors present a robust cohort of 100 cases, I wonder whether any statistical analysis could be added—particularly regarding which findings may be more sensitive or characteristic for KO or EACC. (I understand that this may be limited by space and formatting considerations, so this suggestion is only if feasible.)

Incorporating such an analysis, along with representative endoscopic and CT images as noted above, would substantially increase the educational value of the manuscript, especially for early-career otologists.

I selected “N/A” for the question “Is the research design appropriate?” in the reviewer form.

This choice does not imply that the study design is inappropriate. Rather, based on the title and manuscript, the 100 cases registered between 2021 and 2025 appear to reflect the experience of a single, experienced ENT specialist.

If this is the case, it may mean that cases managed by other physicians were not included, which places the study more in the category of a descriptive single-clinician case series. 

Therefore, I felt that the standard criteria for evaluating study design were not fully applicable. This note is purely for clarification and does not reflect a negative recommendation.

Author Response

Dear reviewer,

Thank you for your comments aiming at improving our manuscript.

  1. On the clinical findings and imaging

From the overall content, the authors appear to suggest that among patients presenting with aural fullness, cerumen impaction is the most common cause, followed by keratosis obturans (KO) and external auditory canal cholesteatoma (EACC), and that distinguishing KO from EACC is clinically important.

In lines 95–96, the authors state that “All participants underwent a comprehensive otologic examination using either otomicroscopy or oto-microendoscopy.

Do the authors have photographic documentation of typical otoscopic findings for KO and EACC? Demonstrating the similarities in external auditory canal appearance between these conditions—while also highlighting the differences, such as the pattern of bleeding after intervention noted in lines 123–124—would be extremely educational for less-experienced clinicians.

If available, representative images comparing KO and EACC would greatly strengthen the paper. Additionally, comparative CT images illustrating the key radiologic distinctions would be highly beneficial.

Answer:

A limitation of this retrospective case series is the lack of comprehensive visual documentation.

Otoscopic photographic or video recording was not consistently available for all cases, as the pathology was managed in outpatient clinics that were not universally equipped with dedicated documentation systems, in contrast to our operating room facilities.

Furthermore, Computed Tomography (CT) imaging was selectively indicated only for patients diagnosed with external auditory canal cholesteatoma (EACC) demonstrating invasion of adjacent structures. Consequently, the available CT documentation is restricted exclusively to this subset of invasive EACC cases, with the physical images currently remaining in the patients' possession.

  1. On patient background factors (lines 93–95)

The manuscript notes that 70 patients were current or past smokers, 15 had chronic rhinosinusitis with or without nasal polyposis, and 8 had atopic dermatitis.

Did any of these factors show a potential association with the clinical presentation or diagnosis?

If so, it may be helpful to comment briefly on their possible implications in the Discussion section. If feasible, providing some statistical analysis related to these background variables could further enhance the rigor of the manuscript.

Answer:

In the present series, we also mentioned that some patients had common inflammatory pathologies or exposure to environmental factors—specifically tobacco smoke exposure, chronic rhinosinusitis with or without nasal polyposis, and atopic dermatitis. To date, the literature does not demonstrate a clear or consistent relationship between these factors and either condition. Tobacco smoke and chronic rhinosinusitis primarily affect the mucosa of the upper airway and middle ear and have no established pathophysiologic impact on the keratinizing epithelium of the external auditory canal. Atopic dermatitis, although theoretically relevant because it is characterized by altered epithelial barrier function and abnormal keratinization, has not been shown in clinical studies to increase the risk of KO or EACC. At most, atopic dermatitis may represent a potential but unproven contributor to local epithelial dysregulation. Overall, our findings align with existing evidence indicating that none of these factors appear to meaningfully influence the development or clinical expression of KO or EACC, both of which remain primarily localized disorders of keratin accumulation and canal wall inflammation.

Although there could be an association with atopic dermatitis, we are unable to confirm it within the context of our study and therefore we prefer not to introduce unsubstantiated hypotheses in the discussion. Some patients in our cohort presented with both atopic dermatitis and chronic rhinosinusitis, with or without nasal polyposis, raising the possibility that chronic rhinosinusitis may act as a confounding factor and that atopic dermatitis might still play a role. However, as previously mentioned, we do not have sufficient evidence to demonstrate a genuine association between these conditions and the clinical presentation or diagnosis of keratosis obturans or external auditory canal cholesteatoma. Nevertheless, we chose to report these observations because they may be useful for future investigations by our group and by other authors, and we thank you for this insightful comment 

  1. On Table 1 and potential statistical analysis

Table 1 is very informative in summarizing the clinical signs associated with cerumen impaction, KO, and EACC. As the authors present a robust cohort of 100 cases, I wonder whether any statistical analysis could be added—particularly regarding which findings may be more sensitive or characteristic for KO or EACC. (I understand that this may be limited by space and formatting considerations, so this suggestion is only if feasible.)

Incorporating such an analysis, along with representative endoscopic and CT images as noted above, would substantially increase the educational value of the manuscript, especially for early-career otologists.

 I selected “N/A” for the question “Is the research design appropriate?” in the reviewer form.

This choice does not imply that the study design is inappropriate. Rather, based on the title and manuscript, the 100 cases registered between 2021 and 2025 appear to reflect the experience of a single, experienced ENT specialist.

If this is the case, it may mean that cases managed by other physicians were not included, which places the study more in the category of a descriptive single-clinician case series. 

Therefore, I felt that the standard criteria for evaluating study design were not fully applicable. This note is purely for clarification and does not reflect a negative recommendation.

Answer:

 Thank you again for your insightful comment!

However, as you rightly observed, given that this study is a descriptive retrospective case series, conventional criteria for statistical analysis and formal study appraisal are not fully applicable. Due to this reason, title has been changed to reflect more accurately the design of our study.

Round 2

Reviewer 1 Report

Comments and Suggestions for Authors

Thank you for submitting the revised manuscript promptly. Recommend including a table showing the number of cases presenting with cerumen vs KO vs EAC cholesteatoma. Also, discuss the management offered to patients with KO and EAC cholesteatoma, what was ultimately done and how did the procedure differ in these two pathologies.  It is insufficient to state " surgical intervention was proposed to these patients" -line 147

The revision was prepared hastily and there are several typos. For example,

Line 102: (es eustachian tube dysfunction....

Line 124: toileting,

Line 187: management reuirements 

Please re-submit after careful revision of spellings, typos and grammar.

Comments on the Quality of English Language

The revised manuscript is prepared hastily. 

English-language editing and careful revision of grammar and typos are necessary. 

Recommend including a table showing the number of cases presenting with cerumen vs KO vs EAC cholesteatoma. Also, discuss the management offered to patients with KO and EAC cholesteatoma, what was ultimately done and how did the procedure differ in these two pathologies.  It is insufficient to state " surgical intervention was proposed to these patients" -line 147

Author Response

We thank the reviewer for the thoughtful and constructive comments.

1. Addition of a table with case distribution
As suggested, we have now included a dedicated table (Table 1) clearly summarizing the number of patients diagnosed with cerumen impaction, keratosis obturans (KO), and external auditory canal cholesteatoma (EACC). This table is positioned at the end of the Results section.

2. Clarification of management in KO and EACC
We expanded the Discussion and Results sections to clarify in detail the conservative management performed for KO (multiple outpatient toilettage sessions) and the attempted outpatient debridement for EACC.

3. Clarification regarding surgery
We added a specific statement explaining that although surgery was recommended, none of the three EACC patients had undergone surgery at the time of manuscript preparation, which is why we did not elaborate on surgical techniques. This study focuses on the outpatient diagnosis and conservative management, not operative treatment.

4. Correction of typos and grammar
We carefully revised the entire manuscript to correct all typographical errors (e.g., “es,” “toileting,” “reuirements”) and improve grammar, clarity, and consistency.

We thank the reviewer once again for the valuable suggestions, which have strengthened the clarity and quality of the manuscript.